# Antifungal Nano-Therapy in Veterinary Medicine: Current Status and Future Prospects

**DOI:** 10.3390/jof7070494

**Published:** 2021-06-22

**Authors:** Mousa A. Alghuthaymi, Atef A. Hassan, Anu Kalia, Rasha M. H. Sayed El Ahl, Ahmed A. M. El Hamaky, Patrik Oleksak, Kamil Kuca, Kamel A. Abd-Elsalam

**Affiliations:** 1Biology Department, Science and Humanities College, Shaqra University, Alquwayiyah 19245, Saudi Arabia; malghuthaymi@su.edu.sa; 2Department of Mycology, Animal Health Research Institute (AHRI), Agriculture Research Center (ARC), 12611 Giza, Egypt; atefhassan2000@yahoo.com (A.A.H.); rasha_hamza2005@hotmail.com (R.M.H.S.E.A.); ahmed_elhamaky@yahoo.com (A.A.M.E.H.); 3Electron Microscopy and Nanoscience Laboratory, Department of Soil Science, College of Agriculture, Punjab Agricultural University, Ludhiana 141004, India; 4Department of Chemistry, Faculty of Science, University of Hradec Kralove, 50003 Hradec Kralove, Czech Republic; patrik.oleksak@uhk.cz; 5Plant Pathology Research Institute, Agricultural Research Center (ARC), 9-Gamaa St., 12619 Giza, Egypt

**Keywords:** nanoantifungal, mycotoxin degradation, theragnostic, veterinary

## Abstract

The global recognition for the potential of nanoproducts and processes in human biomedicine has given impetus for the development of novel strategies for rapid, reliable, and proficient diagnosis, prevention, and control of animal diseases. Nanomaterials exhibit significant antifungal and antimycotoxin activities against mycosis and mycotoxicosis disorders in animals, as evidenced through reports published over the recent decade and more. These nanoantifungals can be potentially utilized for the development of a variety of products of pharmaceutical and biomedical significance including the nano-scale vaccines, adjuvants, anticancer and gene therapy systems, farm disinfectants, animal husbandry, and nutritional products. This review will provide details on the therapeutic and preventative aspects of nanoantifungals against diverse fungal and mycotoxin-related diseases in animals. The predominant mechanisms of action of these nanoantifungals and their potential as antifungal and cytotoxicity-causing agents will also be illustrated. Also, the other theragnostic applications of nanoantifungals in veterinary medicine will be identified.

## 1. Introduction

Fungal diseases are manifested as active infections and/or secretion of mycotoxins on growth of fungi in different tissues of animals. The specific fungal disorders include bovine mastitis, fungal diarrhea in calve, respiratory disorders, superficial, subcutaneous, and systemic infections and mycotoxicosis [1,2]. The variability of the extent of serious public health risk effects of fungal infections in livestock and other domesticated animals spans carcinogenic, nephrotoxic, and hepatotoxic effects following their consumption in the contaminated grains/animal food products [1,3]. Animal production holds considerable economic importance for humans, particularly in low-income countries [4]. Hence, published literature searches included different studies which evaluated the extent of incidences of fungal diseases claiming morbidity and mortality in animals of economic importance besides diverse techniques that can be followed to control the growth of fungal pathogens and secretion of mycotoxins [1,5]. The global prevalence of mycosis and mycotoxicosis related diseases in livestock is about 25%. The traditional treatment procedures including the use of amphotericin (AmB) have been evaluated to be relatively ineffective in most cases due to reactivation of latent fungal infections post medication treatment [2,6]. Likewise, the treatment of fungal infections with azoles (such as fluconazole, voriconazole and itraconazole) may lead to the emergence of resistant fungal pathogens due to excessive and frequent use [7,8]. To solve these issues of fungal disorders in animals, the search for novel effective nanotechnology-enabled antifungals has gained impetus. Further, dual benefits can be reaped through use of nanomaterials for both therapy and diagnosis of disease pathogens separately and for developing conjugate systems for simultaneous diagnosis and targeted release of the therapeutic agent, theragnostic systems [9]. Moreover, novel nano-based disease diagnosis and therapeutic systems have been developed for effective treatment of different animal diseases caused by fungal, parasitic and viral pathogens [3,5]. The antifungal nanomaterials can be applied for the diagnosis of the problems related to reproductive system of the animals [10] and for the protection of the physiological activities of animal genital organs and secretions [11,12]. Also, nanomaterials can be utilized to generate effective vaccines [13]. Nanomaterials can exhibit improved killing or inhibitory activity on fungal pathogens at lower doses and can also be utilized as drug delivery vehicles to help in targeted delivery of drugs [10]. Besides, novel formulations of antifungals or new devices that increase the likelihood of the medication being administered to the site of infection tend to be important in order to boost drug efficacy [14,15]. Therefore, the aim of the present review was to investigate the types of nanoantifungals and their applications in animal health. Also, their uses for mycotoxin degradation in animal feeds, and their therapeutic and preventive aspects were illustrated. Moreover, the mechanisms of nanoantifungal actions, toxicity, and ways to overcome the suspected toxicity will also be discussed.

## 2. Nanoantifungals: Diversity and Relevance for Applications in Veterinary Medicine

Metal/metal oxides and their nanocomposites such as zinc, silver, selenium, copper-chitosan nanocomposite and other nanomaterials exhibit prominent fungicidal activity compared to their bulk counterparts [9,16]. These antifungal nanomaterials can be categorized into various forms according to their chemical sources and morphology [17,18]. A variety of nano-antifungals have been developed to cure different fungal diseases in animals and human beings (Figure 1).

### 2.1. Categories of Nanoantifungals in Veterinary Medicine on Basis of Their Chemical Origin, and Structure

#### 2.1.1. Organic Synthetic and Natural Polymeric NPs

A diversity of nanoparticles and nanoscale products can be developed from synthetic and natural polymers. These NPs are formed from natural and synthetic materials including saccharides and their derivatives such as chitosan, lipids and other biomolecules. A huge variability in the size of these nanomaterials has been reported, with size dimensions spanning from 0.5 to 100 nm. These nanomaterials have high loading/conjugating capacities and have also been used for the development of hydrogel nanoformulations, particularly for the sugars and their derivatives [17]. Synthetic polymeric NPs can be composed of amphiphilic polymers such as caprolactone or PLGA which form a hydrophobic core that facilitates the transportation of hydrophobic drugs encrusted with a water-soluble coat [19]. These NPs have been used for transportation and delivery of drugs with low water-solubility such as amphotericin [20].

Solid lipid nanoparticles developed from a variety of lipids are upcoming drug delivery vehicles which exhibit great potential for lipophilic anti-cancer drugs. These can be easily combined with other materials to induce improved humoral antibody dependent immunity in the animals [21], besides their role in gene therapy by development of nucleic acid-based conjugates [22,23]. The oral, skin and parental routes of solid lipid NPs application are more effective in drug delivery and are highly absorbed [23].

Another category of polymeric nanoparticles includes the most popular forms called liposomes. These are non-toxic PEGylated NPs which are comprised of a two lipid (bilayer) cover shell having high solubility for fatty (hydrophobic) drugs. The first layer of the liposome is coated with a PEG layer to prevent any immune response towards the particles [17,24]. However, due to their vulnerability to get digested in the alimentary canal leading to loss of function, these nanoformulations are preferentially administered through parental and topical routes. Conjugating liposomes with biologically active antibodies can be useful for cancer cell treatment [23]. Further, liposomal formulations of dead pathogens can be utilized to develop vaccines [25]. The liposomes can also be conjugated with DNA to develop DNA vaccines [17]. Furthermore, the liposomes enable drug delivery and diffusion to targeted cell sites within the organism (Figure 1). Despite these benefits and potentially useful activities, these formulations are prone to changes during storage and also the encapsulated compounds may exhibit rapid destruction of their content on account of oxidation processes [26].

Similar to liposomes are the polymeric micelles with one basic difference from the former type that the latter are formed from exfoliated lipid bilayers and thus exhibit great potential to encapsulate lipophilic drugs. Therefore, micelles are hydrophobic core surrounded by a hydrophilic coat which increases their solubility in water [23].

Nano-cochleates are a specialized category of sub-micron to nanoscale solid particulate lipid-based drug carriers [27] which can be derived by the fusion of liposomes with metal cations and involve spiral rolling of continuous lipid bilayer [28,29]. These carriers can be efficiently loaded with both hydrophobic as well as hydrophilic drugs ensuring higher protection from gastrointestinal degradation of anti-fungal drugs particularly Amphotericin and thus enabling oral administration [30,31].

Synthetic polymeric nanoparticles primarily including dendrimers are derivatives of long-chain branched polymers such as polyamidoamines. Similar to micelle nanoparticles, dendrimers are water-soluble, exhibit high biological activities and possess comparatively a much smaller size than the other polymeric NPs discussed so far [26,32]. These attributes of the dendrimers do not allow stimulation of the immune response after parental administration. Dendrimers can be combined with drugs to improve their efficiency for treatment of a variety of animal disorders [26]. The dendrimer formulations have been successfully used for effective cancer treatments and may showcase multiplexed functions including detection of the tumor cells, entry through the cell membrane, targeted release of the conjugated anticancer drugs in the cytoplasm and finally the destruction and death of cancerous cells [23]. Dendrimers can also conjugate with the lipids of the cell membranes and this can create wide pores in the membrane that potentiate improved entrance of the drug containing dendrimer nanoparticles for targeted delivery leading to higher cell death rates [26].

#### 2.1.2. Nanoemulsions

These are aqueous mixtures of oil or other hydrophobic components prepared by addition of oil to water overlaid by non-chemical surfactants [1,16,33,34]. The micelle size in the prepared nanoemulsions may vary from 0.5 to >500 nm. Nanoemulsions exhibit significantly high antifungal, bactericidal and virucidal activities. It may be attributed to greater adherence of the oil droplets on the surface of the microbial cells which facilitates the entrance of drugs to the cell [5,17,35].

#### 2.1.3. Inorganic Metal/Non-Metal Nanomaterials

These NPs are the first-choice nanomaterials to be used as nanoantifungals due to their low cost, easy application, eco-friendly characteristics and wide viability [1,5]. They exhibit potential as antifungals [29,36,37], besides the other biomedical benefits [24]. These nanomaterials may have individual particle size dimensions ranging from 1 to 100 nm with aggregate sizes have a higher size range.

##### Magnetic Iron Oxide Nanoparticles

Magnetic FeO NPs mainly consist of iron core (Fe_3_O_4_ or Fe_2_O_3_) particles which have been used in several studies as significant antifungals against mycotoxigenic molds [34]. Drug delivery, heat therapy and imaging are other beneficial uses of iron core particles [23,38,39]. Moreover, the iron core can be conjugated with fluorescent shells and drugs or antibodies against targeted cancer cells [24]. Further, surface functionalization of these NPs by polyethylene glycol (PEG) can potentially help to prevent elicitation of the immune response.

##### Semiconductor Quantum Dots

Zinc selenide/telluride/sulphide quantum dots exhibit substantial antifungal potential [39]. QDs are core-shell aqueous materials which exhibit conjugation with drugs or other biological materials including nucleic acids (DNA/RNA), proteins and other biomolecules [17]. The biomolecule conjugated QDs have specific use for detection and diagnosis of diseases or their causative pathogens [23]. Further, QDs find peculiar applications for improved imaging and genetic analysis by observing cell activities under disease conditions, and targeted drug delivery [40].

##### Silicate Nanomaterials

These nanomaterials are comparatively biosafe, and do not exhibit high reactivities. Further, the silicate nanomaterials possess diverse morphologies spanning over different particle shapes and sizes which can be easily modified [23]. These silicate nanomaterials are also amenable to functionalization, and other coating treatments. Nanoshells are a specific class of silica nanomaterials which involve a thin metallic coating of the glass core [41]. These nanoparticles have been utilized for the diagnosis of the tumorous tissues [23,42] and simultaneous therapeutics applications [42,43,44,45].

#### 2.1.4. Carbon Nanomaterials

Carbon nanomaterials have significant antifungal and antimycotoxin potential [3,5]. The carbon atom contents enable the destruction of pathogen cell walls [40]. These nanomaterials are insoluble in water and do not get digested in the alimentary tract or get excreted on oral administration [26]. The SE nanomaterials pass through the cell membranes of targeted cells to reach to the cytoplasm of the pathogens or cancer cells causing multiplexed damage resulting in the cell death [46]. Besides, buckyballs can ameliorate pH levels which help in drug delivery to targeted tissues [47]; gene therapy and DNA delivery [48].

#### 2.1.5. Nanobubbles

These are gas core particles suspended in aqueous medium having general size dimensions ranging from 70–120 nm and function as carriers of gas molecules [49]. The nanobubbles are different from the other types of nanoparticles or nanoemulsions as these contain a shell comprised of polymer, phospholipids, proteins or anti-cancer therapeutic agent encasing a gas (generally oxygen) [50,51,52]. These nanomaterials are finding useful applications in diagnosis and targeted delivery of anticancer drugs [49,53].

#### 2.1.6. Nanovaccines and Nanoadjuvants

Today, there are progressive advances in the application of nanotechnology for the production of vaccines. Nanovaccine formulations effectively activate the humoral immunity by a slow elaboration of antigens and thereby elevating the usefulness of vaccination [17,54]. These can be targeted to lymph tissues which significantly enhances the vaccine activities [55]. Nanomaterials conjugated with antibodies and other biological molecules can be used for the quick detection of pathogens and for effective treatment of the diseases caused by them [39]. However, the nanomaterials possess excellent adjuvant properties as these can bind to a variety of antigens/proteins of pathogenic origin to obtain nano-vaccines thereby replacing the use of the adjuvant material [55]. Different forms of nanomaterials used in animal antifungal nanotherapy was shown in Figure 2.

## 3. Applications of Nanoantifungals in Veterinary Medicine

### 3.1. Therapeutic and Preventive Aspects of Nanomaterials

#### 3.1.1. Metal/Metal Oxide/Non-Metal Oxide NPs and their Hybrids as Nanoantifungal Agents

The use of nanomaterials as antifungal agents is an established attribute. The nanomaterials that exhibit antifungal potentials have been evaluated in several studies with primary inhibitory impact on the vegetative growth of the fungal mycelia. The noble metal nanoparticles including the silver and gold nanoparticles possess potent antifungal properties. Nasar et al. [56] have evaluated the broad antimicrobial activity of AgNPs against human pathogenic bacteria (*Escherichia coli*, *Klebsiella pneumonia*, and *Bacillus subtilis*), and common fungal pathogen *Aspergillus niger.* The AgNPs have been found to be effective antifungals against dermal infections [57] Moreover, AgNPs can remove the human oral microbial infections caused by *S. aureus* and *C. albicans* [58], and *C. albicans*, and *Trichophyton mentagrophytes* infections in buffaloes [59]. The nanosized silver can inhibit the growth of *Fusaium* sp. at very low concentrations (<100 ppm) [16,60] and led to decreased mycotoxin production [1,61]. Also, Kischkel et al. [37] observed the antifungal activity of the AgNPs against *C. albicans*, *F. oxysporum* and *M. canis*.

Abd-Elsalam et al. [62] have discussed the fungal growth inhibitory potential of a variety of metal oxide NPs. Among the metal oxide NPs, the most promising candidates are zinc oxide NPs which inhibited the *Candida albicans* growth at very low concentrations of 1.013–296.0 μg/mL [63]. The shape and size of ZnO NPs has been an important characteristic that decides for the extent of the antifungal activity. Flower-shaped ZnO nanostructures inhibited the development of *Aspergillus flavus* and aflatoxin production at concentrations below 5 mM [64]. The next metal oxide NPs showing considerable antimicrobial potential are the iron oxide NPs. A study on magnetic NPs (Fe_2_O_3_ NPs) described the antifungal activity against *A. flavus* and prevention of the aflatoxin production [38]. While, Mouhamed et al. [65] documented the inhibitory effect of iron oxide NPs on ochratoxigenic *Aspergillus* sp. Moreover, Abd El-Tawab et al. [66] have detected the growth inhibitory properties of Fe_2_O_3_ NPs against causative pathogens of bovine skin diseases (*Trichophyton verrucosum*, *T. mentagrophytes*, and *Dermatophilus* sp.).

The coating or surface functionalization of the metal/metal oxide nanoparticles can further improve their antimicrobial properties. The chitosan NPs derived from deacetylated derivative of chitin can prevent growth of *Fusarium* sp., *Rhizopus* sp. and *Aspergillus niger* and thus can be used as an alternative to chemical pesticides [67]. Further, chitosan NPs have also been observed to inhibit fish pathogens under in vitro conditions [68]. Chitosan polymers can also be utilized to develop surface coatings on metal oxide NPs to improve their interactions and passage through the biological membranes. Recently, Abd-Elsalam et al. [69] have detected significant antifungal activity of CuNPs singly and in combination with chitosan against mycotoxigenic fungi, which also led to the prevention of aflatoxin production. The use of an acrylic resin reinforced with ZnONPs and Ag NPs can inhibit the growth of *Candida albicans* [70].

Nowadays, combinations of nanomaterials with beneficial biological active compounds are used to produce nanocomposites of significant use for animal health [1]. The conjugation and overlay of nanomaterials by other biological molecules are related to their chemical properties and used in detection of pathogens inside the body [71]. In this respect, Hassan et al. [1,5] have reported that the conjugation of metals nanomaterials with natural oils significantly improved the antifungal activity. They have detected that the composites of AgNPs, ZnONPs, and essential oils can effectively prevent the growth of fungal and bacterial pathogens. Hybrids of Ag NPs/essential oil were employed in therapy of bovine skin and udder infections [5,72] and carbon NPs [73]. Hassan et al. [5,16,74] have reported the efficient conjugation of ZnNPs and AgNPs with cinnamon and olive oils for use at low safe doses for inhibition of growth of toxigenic *A. flavus* and *E. coli* and production of respective toxins, whereas, Wang et al. [75] successfully detected that the hybrid of Au NPs with antibodies help in immune-chromatographic exploration and diagnosis of toxic AFM_1_ in milk. Similar activities were obtained for QDs to observe events and activities of body cells that were found to be better than the use of traditional dyes and this helped for release of drug to the required site of infection [9,76].

Nanoparticles can also be conjugated with known standard antifungal agents or other molecules where these NPs function as nanovehicles for better delivery of the antifungal therapeutic agents at the targeted site. Therefore, common antifungal agents can also be conjugated on the metal or metal oxide NPs to enhance their antifungal activities. Kischkel et al. [29] have illustrated the potentials of different types of nanoantifungals for the treatment of mycosis caused by *Candida* sp. and *Aspergillus* sp. Hamad et al. [77] have developed a gold nanorod-fluconazole nanoconjugate which exhibited significantly high antifungal activity (9 to 12-fold) against *C. albicans* compared to either component alone, whereas, Huang et al. [36] reported the possibility of using AgNPs as antifungals singly or in conjugation with epoxiconazole (8:2 and 9:1), respectively. The concentration of AgNPs required to suppress the growth of 50% of the fungal colony was 170.20 μg/mL. The combination of AgNPs with fluconazole and florfenicol produced more antimicrobial potential against the causes of animal diseases than their single forms [5].

Inorganic mesoporous silica nanoparticles (MSNs) can also act as nanocarriers for drug delivery to target affected cells inside the body [78]. Functionalized silica NPs can be tethered to drug molecules or they can also adsorb or sequester the drug compound on the surface or inside the nanopores thereby elevating their delivery to the target organs [79,80]. Kanugala et al. [81] have developed phenazine-1-carboxamide-functionalized MSN-based antimicrobial biomaterial surfaces to prevent the formation of bioflms on medical implants. The developed MSNs exhibited superior anti-Candidal activity besides polymicrobial antibiofilm potential. Silica NPs can also be used for the development of topical cream formulations to treat skin fungal infections. Montazeri et al. [82] have synthesized and evaluated an aminopropyl functionalized MSN-econazole topical cream formulation against *Candida albicans* skin infections and observed improved antifungal activity at lower concentrations of the loaded drug.

#### 3.1.2. Polymer Nanoparticles for Antifungal Drug Delivery

Recent drug and vaccine delivery strategies in biomedical research advocate the use of nanomaterials for successful delivery of drugs to targeted cells and tissues [10]. These strategies are beneficial as they can ensure the delivery of drugs to target tissues resulting in a decrease in the amount and required doses for the treatment of diseases. The most promising nanodelivery agents for drugs can be the polymer nanoparticles encapsulating antifungal drugs. In these respects, chitosan (CS) NPs which themselves possess considerable antifungal potential can be used for the delivery of the antibiotic drugs [68]. The encapsulation of antifungal drugs or development of their formulation as nanoemulsions can improve their action potential. Deaguero et al. [83] observed that nanoencapsulation of miconazole in cholesterol/sodium oleate vesicles have significant antifungal activity against several fungal pathogens. Siopi et al. [84] have reported that the liposome-encapsulated amphotericin B possess significant therapeutic potential against mycotic respiratory infections in animals caused by *A. fumigatus*.

Drug molecules can also be nanoformulated as nanomicelles comprised of a hydrophobic core and hydrophilic shell which improves the water solubility and therefore bioavailability of the hydrophobic drugs [85]. Further, these nanosystems can be used for the targeted delivery of the drug [86], treatment of cancer in animals [87] and to ensure drug delivery without stimulation of immunity [88].

#### 3.1.3. Carbon Nanomaterials as Nano-Antifungals

Different forms of carbon-based nanomaterials also exhibit antimicrobial activity against bacterial and fungal pathogens causing diarrhoea [3]. These nanomaterials can inhibit the growth of *E. coli* and mycotoxigenic fungi [89]. Furthermore, conjugation of sugars with CNTs improve the ability to affect the viability of *C. albicans*, *A. flavus* [5]. Several benefits of nanoantifungal applications were detected, as illustrated in Figure 3.

#### 3.1.4. Nanocomposites for Antifungal Drug Delivery Agents

The nanocomposites of natural materials such as carbohydrates and proteins with polymers have the ability of effectively releasing of these materials at targeted sites [40]. However, the modified CS NPs are capable of drug delivery to diseased tissues at lower doses than traditional chemical cancer therapy. They can be used as an adjuvant for effective animal vaccination against infections. The parental administration of nanoshells comprised of silica core attached with metals NPs and drugs in animals can be useful to search and can be directed to target cancer cells [90].

The nanocomposites qualify quite uniquely considering the non-stimulation of the elaborate immune response aspect [91]. The embedding or encapsulation of the drug in polymer blend-based nanocomposite also improves its antifungal potential. Terbinafine hydrochloride was introduced into the polycaprolactone (PCL)/gelatin nanofibers generated by the hydrothermal method [92]. The resulting wound dressings were tested for their antifungal potential.The researchers were successful in inhibiting the *T. mentagrophytes* and *Aspergillus fumigatus* due to slow release of the embedded drug molecules from the nanocomposite fibers over time [92].

### 3.2. Antifungal Nanomaterials for Management of Mycotoxins in Animal Feeds

The morbidity and mortality caused due to global incidences of mycotoxicosis in animal and poultry industry have serious economic repercussions affecting the productivity [1,3,5,93]. Recently, it was reported that ZnO NPs and Fe_2_O_3_ NPs have antifungal activity against ochratoxigenic *Aspergillus* and hence prevent mycotoxin synthesis [57,65]. The supplementation of Zn NPs in aflatoxicated feed of rats and rabbits resulted in the removal of the carcinogenicity of aflatoxins on the kidney and liver [72,94]. The ZnNPs and AgNPs can inhibit the growth of *Fusarium poae* and prevent formation of trichothecenes mycotoxin [16]. The Ag NPs can eliminate aflatoxins in chickens feed [95,96]. Biosynthesized spherical SeNPs produced by *Saccharomyces cerevisiae* and originated from selenous acid and sodium sulfite were able to inhibit pathogenic saprophytes, yeasts, and dermatophytes [97]. Fadl et al. [98] have reported that CuNPs inhibit ochratoxigenic molds and prevent ochratoxin production in a fish feed.

Apart from the metal/metal oxide nanoparticles, carbon nanomaterials, particularly carbon nanodiamonds, can ameliorate the adverse effects of mycotoxins by the process of immobilization of the mycotoxins [99]. While, another report by Hassan et al. [3] detected the activity of CNTs in suppression the toxicity of *A. flavus* at a concentration of 125 μg/mL. Therefore, the primary modus operandi for the anti-mycotoxigenic effects of both nanomaterials and nanocomposites such as iron NPs [100] and MgO-SiO_2_ nanocomposite [101] is through adsorption of the mycotoxins.

Nanohybrids such as polyene-functionalized magnetic NPs possess enhanced antifungal activity against opportunistic oral fungal pathogens such as *Candida* sp. [102]. A miconazole nanocarrier (MCZ) based on iron oxide nanoparticles (IONPs) functionalized with CS was prepared, characterized and screened for antifungal activity against *Candida albicans* and *Candida glabrata* biofilms. A nanocarrier with less than 50 nm dimeter presenting MIC values lower than those observed for high diameter and showed synergism against *C. albicans* [103]. Similarly, nanoformulations of known antifungal agents can improve the action spectrum of these agents and enhance the antibiofilm potential. The antimicrobial and antibiofilm effects of a colloidal nanocarrier for chlorhexidine (CHX) on yeast and bacteria such as *Candida glabrata* and *Enterococcus faecalis* were evaluated. The CHX nanocarrier has an excellent ability for the management of oral diseases linked to *C. glabrata* and *E. faecalis* [104].

The nanocomposites derived from metal oxide and carbon nanomaterials have also been evaluated for anti-mycotoxin properties. A magnetic carbon nanocomposite derived from bagasse was observed to degrade AFB1 [105], while graphene oxide nanocomposites caused a reduction in the occurrence of three prominent *Fusarium* toxins i.e., ZEA, FB, and deoxynivalenol [106] and modified halloysite nanotubes [96,107]. Also, detoxification of AFB1 by a magnetic graphene oxide nanocomposite has been reported by Ji and Xie [108]. González-Jartín et al. [109] observed that nanocomposites of carbon, bentonite, and aluminum oxide eliminated up to 87% of the mycotoxins with an adsorption efficiency of 450 µg/g. Chitosan-stabilized selenium nanoparticles have a significant ability to improve the toxic effects of aflatoxicosis in rats [35,110]. Further, the SeNPs exhibited important inhibitory effects on *A. parasiticus, A. ochraceus,* and *Aspergillus nidulans* growth at concentrations varying from 0.1–0.5 mg/L and ameliorate the dysfunction and hepatic apoptosis induced by AFB1 [30]. Chitosan-coated Fe_3_O_4_ particles have been reported to be substantially useful for patulin decontamination with no toxic response or histopathology in treated mice [106,111]. Recently, Hassan et al. have also assessed the efficiency of the copper-CS nanocomposites for the removal of the aflatoxins and ochratoxins in poultry (personal communication).

Conjugating metal oxide nanoparticles with other antimicrobial components such as essential oils, curcumin or ozone can improve the antimycotoxigenic activities. Also, the antimicrobial, anti-aflatoxins, and anti-shigella toxins potentials of nanoemulsion of cinnamon oil and ZnO NPs towards to fungal causes of dysentery in buffaloes were detected [16]. Hassan et al. [72] observed that conjugating ZnO-NPs along with probiotic and curcumin improved the inhibitory activities on mycotoxin producing *Fusarium* sp. besides significantly decreasing their ability for mycotoxin production. The combined application of ZnO NPs, probiotic and curcumin (ZnO NPs (100 µg/mL) + probiotic (0.5%) or curcumin (0.5%)) resulted in complete detoxification of *Fusarium* mycotoxins [72]. Hassan et al. [16] reported alteration in the gene expression profile of the ZnO and essential oil treated *E. coli* and *A. flavus* through RT-PCR studies that helped to elucidate the efficacy of the treatments. When the treatment doses of ZnO NPs, cinnamon oil, and olive oil increased, the AflR and Stx toxin genes expression efficacy, the molecular weight of DNA, and cycle threshold were decreased. The synergistic activity used lower doses of combined form than each alone. Hamza et al. [112] used hybrid β-glucan mannan lipid particles (GMLPs)-humic acid iron nanoparticles (HA-FeNPs) as an AFB1 binder provides a high binding capacity and a safe enhanced mycotoxin binding material.

### 3.3. Cancer Theragnostics

Nanoantifungals and their hybrids have the potential to penetrate cancer cells and accumulate in the cytoplasm of cancer cells leading to damage, followed by an inhibitory action leading to death [111]. Similar properties of nanoparticles can help in early and reliable detection of various types of cancer tissues [112]. Hybrid nanocarrier and natural body sugars-derived NPs can help in carrying and release of drugs as in cases of lung cancer [88].

#### 3.3.1. Cancer Therapeutic Applications

##### Nanoantifungal Agents and Their Hybrids

Zn NPs enable killing of the tumor cells that may also help to preserve the immune cells intact and this activity can be used for both tumor detection and therapy (cancer theranostics) simultaneously [113]. Anticancer drug-functionalized iron or zinc nanoparticles can improve the adherence of the drug-NP conjugate to the target tumor cell and can ensure targeted release of the anticancer drugs to malignant or tumor cells [114,115,116]. Another study detected that magnetic NPs encapsulated with silica can be effectively used as antitumor drugs [117]. Polymeric nanoparticles such as solid lipid nanoparticles and dendrimers are also potent smart nanovehicles ensuring the targeted delivery of drug molecules in cancer tissues. Nanoencapsulation of 5-fluorouracil in solid lipid nanoparticles improved the specific problems associated with rapid metabolism and shorter life time [118]. This nanovehicle therefore improved the use of the 5-fluorouracil for treatment of colorectal cancer conditions. Meena et al. [40] and Hassan et al. [3] detected the major benefits of CNTs fungal and tumor infections treatment. Xie et al. [119] successfully demonstrated the relevancy of a carbon nanoparticle suspension injection for the diagnosis of thyroid carcinoma.

##### Nanocomposites

The combination of nanomaterials and drugs, sugars, proteins and DNA potentiated detection and control of animal tumors [120]. The parental inoculation of nanocomposite of Au NPs and gum Arabic act as fluorescent agents in canine cancer therapy [121]. Osama et al. [76] found a significant ability of liposomes to reach the targeted tumors tissues and effective drugs release. Osama et al. [76] estimated the viability of liposome hybrid NPs in detection and therapy of canine tumors in the spleen. Furthermore, dendrimers, the hybrid nanomaterials have the potentials to be conjugated with biological materials and anticancer drugs and released them in targeted tissues in the body and hence excellent tumor detection and treatment were achieved (Figure 4). The combination of MSNPs/folic acid resulted in possibilities of direction of a drug to tumor cells and hence treatment of tumors in mice [122].

#### 3.3.2. Cancer Diagnosis Applications

The essential role for the correct disease diagnosis involves clear observation of the affected tissues activities via imaging.

##### Nanoantifungal-Based Diagnostic Approaches

The majority of antifungal nanomaterials such as magnetic nanoparticles (MNPs) can be employed in MRI imaging of body tissues [123]. A particular benefits of these NPs is their greater ability to penetrate through the cell membranes and reach blood supply to contrasts of targeted cells as canine stem cells [124]. Superparamagnetic iron oxide nanoparticles functionalized with PEG and ^64^Cu exhibited encouraging PET and MRI imaging properties besides possessing good stability [125]. Packed graphene oxide also possesses significant potential for quick and sensitive detection and treatment of infections [126]. QDs, the semiconductor materials exhibit huge potential for disease diagnostic applications [127,128,129].

##### Nanocomposites

QDs as Co@Cd-Se core-shell nanocomposites and FePt-Zn nanosponges have fluorescence properties that help in imaging biological events [130]. However, the conjugation of QDs with biological materials (AS enzymes, antibodies and DNA) caused markers imaged by a fluorescence signal [131,132]. In addition, QDs are more photostable than traditional chemical dyes which makes them to be appropriate for use in bioimaging [133,134].

### 3.4. Nanoantifungal-Enabled Improved Animal Nutrition, and Breeding

Recent research focused on the use of nanomaterials for improving the efficiency of animal production has gained the attention of veterinary experts. Some relevant aspects include the reports on the supplementation of CuNPs, ZnNPs and SeNPs in chicken feed that elevated their productivity of egg and meat [135,136]. Addition of ZnO NPs to broiler chick feeds resulted in elevation of their health status and growth performance [137,138]. Moreover, multiplexed positive effects of these nanomaterials can be identified such as the fact these materials increased the growth rates, reproductive viability, and meat and egg quality of animals and poultry [4]. Also, the supplementation of coated nanomaterials kept their viability against the worst environmental conditions such as digestive enzymes, light and oxidation [40,76]. Another study on injection of Ag NPs alone or in combination with cysteine/threonine amino acids in chicken embryos increased the formation of breast tissue and also improved the chicken immunity through the immunomodulatory properties of the NPs [139]. Also, semiconductor QDs have been successfully utilized for the detection and imaging of physiological events related to functioning of the spermatozoa and female gametes [140] and imaging of fertilization events in male pig gonadal tissue [128]. QDs have the potential to determine the spermatozoon and oocyte movements, hence significant improvement in animal production occurred [12]. NPs can be used not just imaging for the elucidation of the gamete functions, but also as antibody or lectin conjugated metals for the segregation or fractionation of abnormal sperm from active healthy sperm if the functionalized antibodies can detect the defective sperms [141].

Antifungal nanomaterials showed significant activities for elevation of the efficacy of animal reproduction aspects [9,12]. Nanomaterials can be utilized to improve the life and efficacy of preserved semen specimens. The supplementation of polyethyleneimine or propyltriethoxysilane-functionalized mesoporous silica NPs did not exhibit any negative impact on any sperm activity-related properties [142]. Thus, these NPs may help in preservation of the semen quality during in vitro artificial insemination [142]. Another report showcased an improvement in the fertilization potential of the buffalo sperm on addition of titanium oxide NPs (TiO_2_ NPs, 10 μg/mL) [143].

Administration of NPs of antioxidant compounds or vitamins can improve the ability of the organism to withstand and avoid oxidative stresses. Oral administration of α-tocopherol NPs in equine animals showed significantly improved rates of absorption and α-tocopherol plasma levels because of maintenance of the high oxidative status in race horses undergoing strenuous training [144]. The pig health status can be elevated by supplementation of micellar NPs conjugated with vitamin E to pigs [145]. Nanosized nutrients and vitamins used as a feed additive in feeds and pass through the alimentary tract of an animal to the blood vessels and distributed to different biological tissues cause their significant improvement [146].

## 4. Nanoantifungals: Can These Be the Future Innovations in Veterinary Biomedicine?

### 4.1. Mechanism of Action of Nanomaterials as Antifungal Agents

Nanomaterial possess a range of activities to inhibit the growth and multiplication of fungal-pathogens resulting in cell damage and loss of functions [3,5,35,38]. Nanomaterials exhibit a large surface area compared to the corresponding bulk materials [147]. These materials interact with the various biomolecules in the biological milieu eliciting formation of reactive oxygen species. The action of several nanoantifungals leads to an augmentation of intracellular ROS, an important mediator for exerting antifungal effects. The antifungal activity of nanosilver has been associated with the induction of mitochondrial dysfunctional apoptosis through an increase in oxidative stress via ROS generation especially hydroxyl radicals [148]. The ROS generation is initiated as a response to attachment of antifungal nanomaterial with targeted cells leading to elaboration of O_2_ atom and metal ions [149], whereas, the elaborated O_2_ increases the oxidative stress causing damage of the mitochondria proteins, leading to denaturation and loss of their functions. These potentials of ROS production have been observed on supplementation of C_60_ fullerenes, SWNTs, and QDs [1,5,150].

### 4.2. Cytoxicity Risks of the Use of Nanoantifungal Agents

The continuous awareness about the toxicity risk of nanomaterials to animal and the environment have led to refusals of applications of nanomaterials in animal science by several international authorities [151]. The toxicity of nanomaterials can be affected by a variety of factors such as particle size, dose level, type of animal species and the period of exposure [94] and the physico-chemical characters of the nanomaterials used [152]. Chronic exposure of buffalo sperm to ZnNPs and TiONPs (100 μg/mL) caused several abnormalities resulting in suppression of viability and diminished fertility [143], while, sperm exposure to 100–500 μg/mL of Zn NPs caused their damage and rapid death [153]. Hence, the estimation of safe doses of the used nanomaterial should be investigated in laboratory animals before application to field animals [35,72,94].

Furthermore, upon ingestion of nanomaterials by humans and animals they enter the alimentary tract, reach the circulatory system and are carried over via the liver and spleen [35,154,155], whereas, inhalation and skin exposure to nanomaterials allows for their penetration through the skin tissues and nerve cells [152,155]. Inhalation of TiO_2_ NPs was identified to have an effect in the development of lung cancer [154]. When NPs reach blood vessels, pathological effects occur such as blood clots and disorders in the cardiovascular system functions [156]. Inhalation of low doses of TiO_2_ NPs can cause vascular disorders in rats [157], besides inhalation of single wall and multiwall CNTs [158,159]. We have little knowledge about toxicity and the journey of the nanoparticles in the animal body from the site of administration, passage through absorption, blood vessels, distribution in body tissues and their further journey. Hence, broad toxicological studies are needed before launching commercially viable nanotechnology applications in biomedicine and animal health.

### 4.3. Safety Concerns of Nanoantifungals

There are many challenges related to the potentially toxic effects of nanomaterials. Incorporation of nanomaterials into polymeric hydrogel matrices may reduce the toxicity and improve its efficacy because of sustained and controlled release of the incorporated NPs. The effective delivery of the nanomaterials can be ensured by their functionalization with polymers at low doses to avoid elicitation of the cellular toxicity [1,3,5,160]. Moreover, several benefits of nanomaterials use for improvement in biomedical applications have also been realized. Although, information related to their harmful impacts is not sufficient and special attention is required for identification of their toxicity risk before practical biomedical applications can be approved for use.

## 5. Conclusions and Future Perspectives

Over the past decades, nanotechnology has offered progressive novel advances to improve animal health and production. Today, several nanomaterials are used as nanoantifungals, besides having other benefits such as disease detection, diagnosis and therapy, use of additives to animal feed and their products, and finally food safety. The essential therapeutic and preventive activities of nanoantifungals, particularly the zinc and copper nanomaterials, have been evaluated against a variety of fungal diseases and mycotoxicosis in animals. Also, super paramagnetic iron, semiconductor quantum dots and gold nanoparticles are finding applications for early and sensitive detection followed by detailed prognosis and therapy of cancer. Both inorganic and organic polymeric nanomaterials have also been utilized for targeted delivery of various vaccines, quick and on-site detection of pathogens or their signature protein and other biological molecules. The mechanisms of nanoantifungal activity are related to their ability to penetrate the cell membrane, damage the cytoplasmic contents, leading to loss of function and death of the cells. Therefore, further studies illustrating the cellular toxicity mechanisms that result in oxidative stress and leading to genotoxicity and cancers need a detailed evaluation to manipulate the roles of nanomaterial in animal health. Moreover, the toxicity risks of nanomaterials must be determined before application of nanomaterials in veterinary medicine for safeguarding the health of the animals and their role in animal production.

## Figures and Tables

**Figure 1 jof-07-00494-f001:**
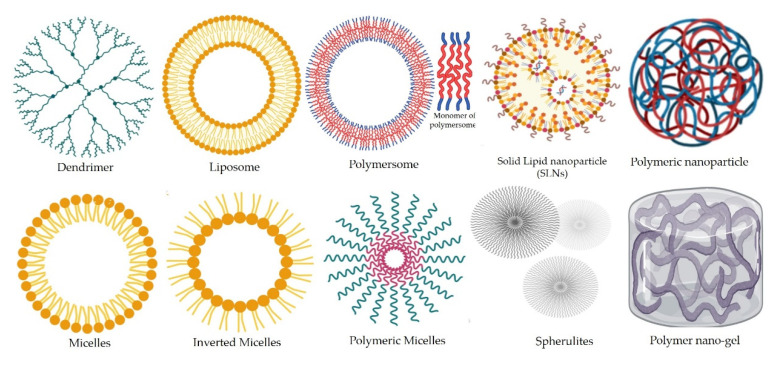
Nanovehicles for effective and smart delivery of therapeutic drugs and other anti-fungal agents.

**Figure 2 jof-07-00494-f002:**
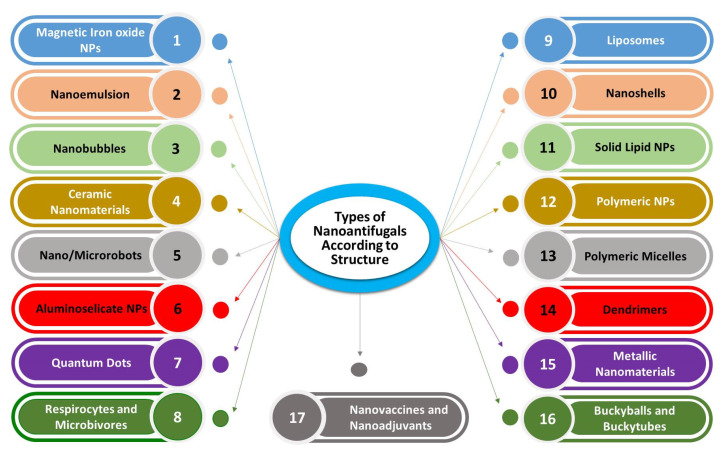
Various types of nano-based materials employed in antifungal nanotherapy in veterinary medicine.

**Figure 3 jof-07-00494-f003:**
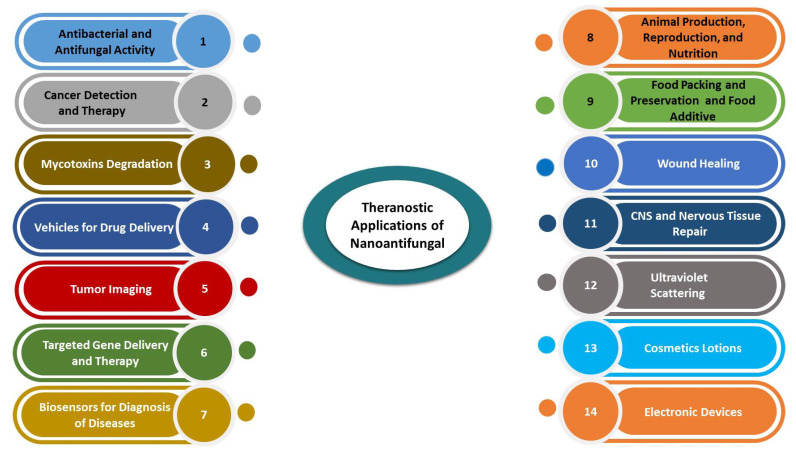
Theragnostic applications of nanoantifungals in animal science.

**Figure 4 jof-07-00494-f004:**
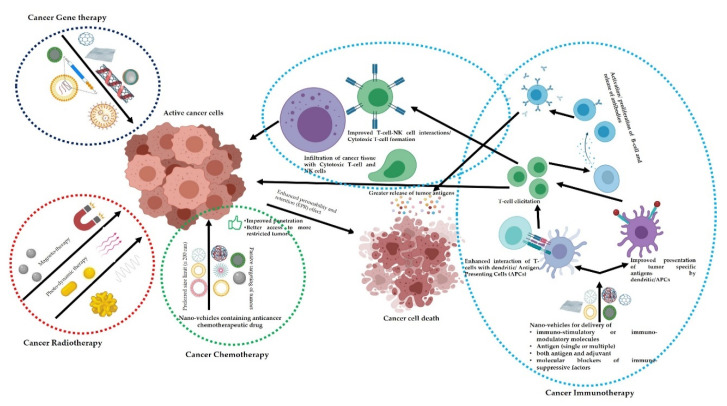
Immuno-modulatory and other functions of nano-antifungals for cancer therapeutics.

## Data Availability

Not applicable.

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
