# Peer review of "Antifungal Nano-Therapy in Veterinary Medicine: Current Status and Future Prospects"

_jof, 2021, doi:10.3390/jof7070494_

Round 1

Reviewer 1 Report

A review entitled “Antifungal Nano-therapy in Veterinary Medicine: Current Status and Future Prospects,” by Alghuthaymi et al. survey nanoantifungals and their use in the development of a variety of pharmaceutical and biomedical products such as nano-scale vaccines, adjuvants, anticancer and gene therapy systems, farm disinfectants, animal husbandry, and nutritional products. Authors emphasize the significance of this issue and its goals in this Review. In general, the paper's text is fluent and understandable. However, the following changes are suggested/recommended to improve the quality of paper as below.

  1. Abstract of the review should be scientifically enriched so that it is easily understood by the reader.
  2. It is necessary to get rid of the confusion. (“Advancement in biomedical applications of nanotechnology derived products and processes have gained” sentence shows confusing “nanotechnology derived products”)
  3. Line 61; -cidal is the suffix, so write the whole word

Line 65; Replace this paragraph “Therefore, the present review was aimed to investigate ……….. animal health by “Therefore, the aim of the present review was to investigate………………….health.

  1. To make subheadings more interesting, the author also provides a range of size using standard literature, for example, a range of size of Nanobubbles etc.
  2. Line 286, Write full form of abbreviation (CS ??? or any other); Put the abbreviation in parentheses after the entire term the first time you use it. After that, you can just use the acronym.
  3. Kindly select standard references belong to ISI/SCI/SCIE for writing (see reference 35, replace it by standard reference). Self-citation is not a problem, but the paper should be standard
  4. Provide page number and volume of “Hassan, A.A.; Howayda, M.E.; Mahmoud, H.H. Effect of Zinc Oxide Nanoparticles on the Growth of Mycotoxigenic Mould. 2013”
  5. Line 98; Introduce a paragraph to explain the meaning of liposomes in a simple way
  6. Conclusion section should be improved with selected and emphasized significant results from the review, as well as your points of view
  7. Finally replace some of the references 35, 55, 62, 64, 71, 72, with the standard one

Author Response

Dear Reviewer,

Sincere thanks for providing the valuable suggestions for the improvement of this manuscript. The point-wise reply to the suggestions/ comments/ queries have been appended below:-

Comment 1: Abstract of the review should be scientifically enriched so that it is easily understood by the reader.

Reply: The abstract has been revised.

Comment 2: It is necessary to get rid of the confusion. (“Advancement in biomedical applications of nanotechnology derived products and processes have gained” sentence shows confusing “nanotechnology derived products”).

Reply: The indicated sentence has been modified.

Comment 3: Line 61; -cidal is the suffix, so write the whole word

Reply: The whole word has been written wherever –cidal suffix appears in the manuscript.

Comment 4: Line 65; Replace this paragraph “Therefore, the present review was aimed to investigate ……….. animal health by “Therefore, the aim of the present review was to investigate………………….health.

Reply: The indicated changes have been incorporated in the revised manuscript.

Comment 5: To make subheadings more interesting, the author also provides a range of size using standard literature, for example, a range of size of Nanobubbles etc.

Reply: The range of size for various nanomaterials has been incorporated in the revised manuscript.

Comment 6: Line 286, Write full form of abbreviation (CS ??? or any other); Put the abbreviation in parentheses after the entire term the first time you use it. After that, you can just use the acronym.

Reply: The full form of the abbreviations have been incorporated.

Comment 7: Kindly select standard references belong to ISI/SCI/SCIE for writing (see reference 35, replace it by standard reference). Self-citation is not a problem, but the paper should be standard.

Reply: The reference 35 has been removed.

Comment 8: Provide page number and volume of “Hassan, A.A.; Howayda, M.E.; Mahmoud, H.H. Effect of Zinc Oxide Nanoparticles on the Growth of Mycotoxigenic Mould. 2013”

Reply: The page number and volume have been incorporated.

Comment 9: Line 98; Introduce a paragraph to explain the meaning of liposomes in a simple way

Reply: The paragraph regarding liposomes has been incorporated.

Comment 10: Conclusion section should be improved with selected and emphasized significant results from the review, as well as your points of view

Reply: The conclusion has been revised as per the suggestions to include specific and significant results.

Comment 11: Finally replace some of the references 35, 55, 62, 64, 71, 72, with the standard one.

Reply: The indicated references have been deleted.

Reviewer 2 Report

introduction

Line 35, author should mention major diseases caused by fungi

Line 55-57, how nanomaterial can be used for diagnosis, give an example

The authors should write about the importance of nanomaterials in decrease the effect of mycotoxicosis in animal foods

In part of application of nanoantifungal, it needs rephrasing, it is very complicated and have mistakes

Line 217, staph aureus does not cause fungal infection

Author Response

Dear Reviewer,

Sincere thanks for providing the valuable suggestions for the improvement of this manuscript. The point-wise reply to the suggestions/ comments/ queries have been appended below:-

Comment 1: In introduction, Line 35, author should mention major diseases caused by fungi.

Reply: The fungal disorders mentioned are in order of their occurrence or reports in animals.

Comment 2: Line 55-57, how nanomaterial can be used for diagnosis, give an example

Reply: The use of magnetic Fe nanoparticles and semiconductor qdots have been useful in disease diagnosis and prognosis through magnetic resonance imaging and fluorescence or bioimaging techniques. The details have been provided in the section 3.3.2.1 in the manuscript.

Comment 3: The authors should write about the importance of nanomaterials in decrease the effect of mycotoxicosis in animal foods

Reply: The role of nanomaterials in curbing mycotoxicosis includes curbing the growth of mycotoxigenic molds and removal of the released mycotoxins by the mechanisms of adsorption/ sequestration which has been provided in the sections 2.1.4, 3.1.1 and 3.2 of the manuscript.

Comment 4: In part of application of nanoantifungal, it needs rephrasing, it is very complicated and have mistakes.

Reply: This section has been subdivided into four subsections including the 3.1 (Therapeutic and preventive aspects of nanomaterials), 3.2 (Antifungal Nanomaterials for management of mycotoxins in animal feeds), 3.3 (Cancer theragnostic), and 3.4 (Nanoantifungal enabled improved animal nutrition, and breeding). The further sub-sub-sections under each section describe the specific aspects of a particular nanomaterial(s). 

Comment 5: Line 217, staph aureus does not cause fungal infection.

Reply: The staph aureus context has been deleted.